# Hand Hygiene, Mask-Wearing Behaviors and Its Associated Factors during the COVID-19 Epidemic: A Cross-Sectional Study among Primary School Students in Wuhan, China

**DOI:** 10.3390/ijerph17082893

**Published:** 2020-04-22

**Authors:** Xuyu Chen, Li Ran, Qing Liu, Qikai Hu, Xueying Du, Xiaodong Tan

**Affiliations:** 1School of Health Sciences, Wuhan University, Wuhan 430071, China; 2017203050018@whu.edu.cn (X.C.); 2019103050005@whu.edu.cn (L.R.); liuqing@whu.edu.cn (Q.L.); xueyingdu@whu.edu.cn (X.D.); 2School of Mathematics and Statistics, Wuhan University, Wuhan 430071, China; wsxgshqk@umich.edu

**Keywords:** hand hygiene, mask-wearing behavior, risk factors, COVID-19, primary school student

## Abstract

Although the emphasis on behaviors of hand-washing and mask-wearing was repeated during the pandemic of Coronavirus Disease 2019 (COVID-19), not everyone paid enough attention to this. A descriptive statistic was used to make sense of the status of hand hygiene and mask-wearing among primary school students in Wuhan, China. A binary logistic regression analysis was conducted to identify the risk factors affecting the behaviors of hand-washing and mask-wearing. *p* < 0.05 (two-sides) was considered as significant at statistics. 42.05% of the primary school students showed a good behavior of hand-washing, while 51.60% had a good behavior of mask-wearing. Gender, grade, out-going history, father’s occupation, mother’s educational background, and the time filling out the survey were significantly associated with hand hygiene, whereas grade, mother’s educational background, and residence were associated with mask-wearing. The behaviors of hand-washing and mask-wearing among primary school students were influenced by gender, grade, and other factors, therefore, parents should make efforts of behavior guidance whereas governments should enlarge medium publicity.

## 1. Introduction

Due to the strong human-to-human transmission power of Coronavirus Disease 2019 (COVID-19), now it has been pandemic over the world. Chinese experience, which treats the infected patients aggressively, on the other hand, protects susceptible populations and cuts off transmission routes, which are proved to be a huge success in the fight against COVID-19.

Both pharmaceutical measures and non-pharmaceutical measures are available against COVID-19. Although pharmaceutical measures are the most highly effective strategy, it takes a lot of time to develop vaccines and antiviral medications, so they cannot control an outbreak caused by a new pathogen in the early stage. Under this circumstance, non-pharmaceutical measures like wearing face masks and washing hands are important to reduce the risk by establishing a barrier to curb the aerosol spread and protect susceptible populations [1,2]. At present, the main measures taken abroad are to maintain hand hygiene and an appropriate social distance, however, wearing masks is also advocated in China. Hand-washing is suitable for all people, but keeping the social distance is only suitable for adults when they go out. For children, home isolation is one of the main measures. Washing hands with soap and wearing a mask as a means of preventing and controlling infectious diseases has the advantages of simple operation, strong sustainability, high health benefits, and good health economic benefits [3,4,5,6]. Evidence from the literature showed that frequent hand-washing would reduce the risk of viral transmission by 55% [7,8]. Masks can purify the air entering the lungs through filtration and display an excellent effect in respiratory infectious disease epidemics. During the outbreak of severe acute respiratory syndrome (SARS), hand-washing and mask-wearing were proved to be effective in blocking viral spread [7].

Given widespread population vulnerability to COVID-19 infection, hand hygiene and face masks are repeatedly emphasized in the whole population [9]. Nevertheless, primary school children, whose personal protection is often overlooked. Until very recently, related studies emphasized the importance of hand-washing and mask-wearing in healthcare workers, instead of primary school children who are also at high risk of COVID-19. To make up the gap and better understand the current situation of hand hygiene and face masks among primary school students, we conducted this study in Wuhan, the hardest-hit area.

## 2. Materials and Methods

### 2.1. Participants

The cross-sectional and quantitative study was conducted from 16 February 2020 to 25 February 2020. A total of 9145 students from 15 primary schools in Wuhan, China participated in the survey. The exclusion criteria of the participants were as follows: (1) Filling time less than 90 s; (2) Repeated filling (IP and basic information are completely consistent); (3) Information of residence or school is missing. 576 substandard questionnaires were excluded, the questionnaire-reclaiming efficiency is 93.70%. Since the non-outgoing students have been living at home, many students have not been wearing masks for nearly a month. Therefore, only the out-going students are included in the section of mask-wearing, but the section of hand hygiene includes all students.

### 2.2. Procedure

The program was initiated by Wuhan Centers for Disease Control (CDC). An official document issued by Wuhan CDC was sent to all primary schools in Wuhan. The online questionnaire was forwarded to the student’s parents by the school teacher and were filled in by the students uniformly, the parents were required to supervise their children and explain the questionnaire. A simple random sampling technique was conducted. The sample size was allocated by considering the population in Wuhan.

### 2.3. Instrument

The structured questionnaire consisted of three parts: (1) Sociodemographic characteristics, including gender, grade, outgoing history, mother and father’s occupation, mother’s and father’s educational background, residence, and time filling out the survey (one month backward). (2) Hand hygiene, with 10 items, including hand washing practice at the critical times, method of hand-washing and correct hand habit [10]. (3) Mask-wearing behavior, with 7 items, including mask-wearing habit, the proper mask type, the proper size of a mask, frequency of changing a mask, knowledge and behavior of mask use (see details in the Appendix A).

### 2.4. Variable Definitions

Independent variables included all sociodemographic characteristics. These variables consisted of: gender (1 = male, 2 = female), grade (1 = grade 1 and grade 2, 2 = grade 3 and grade 4, 3 = grade 5 and grade 6), out-going history (1 = no, 2 = yes), mother’s and father’s occupation (1 = temporary workers or laid-off and unemployed, 2 = professional and technical personnel (doctors, teachers, lawyers, engineers), 3 = general employees (general civil servants, company employees), 4 = managers and leaders of state agencies, enterprises and institutions, 5 = business and service workers, 6 = peasant, 7 = operators of production and transportation equipment, 8 = others), mother’s and father’s educational background (1 = primary school and below, 2 = junior high school, 3 = high school or technical secondary school or secondary vocational technical school, 4 = junior college or undergraduate, 5 = postgraduate and above), residence (1 = Wuhan, 2 = Hubei except for Wuhan, 3 = China except for Hubei) and time filling out the survey (one month backward) (1 = before Wuhan locked-down, 2 = after Wuhan locked-down). The dependent variable is “whether the behavior is good or not”, which is classified according to the total score. The score of mask-wearing behavior more than or equal to 6 points (out of 8 points) and hand hygiene more than or equal to 8 points (out of 10 points) was classified as good (1 = good, 0 = poor).

### 2.5. Statistical Analysis

All statistical analyses were conducted using SPSS version 20.0 (IBM Corp., Armonk, NY, USA). If a normality test was not statistically significant (*p* > 0.05), data were presented by frequencies and percentages. The chi-squared tests were used to compare the good rate in different groups and to choose independent variables for the binary logistic regression analysis. Independent variables with *p* < 0.05 were included in the binary logistic regression analysis (Forward: LR). Finally, variables that had a significant association with good hand hygiene or good mask-wearing behavior were identified on the basis of odds ratio (OR) with 95% confidential interval (CI). *p* < 0.05 (two-sides) was considered as significant at statistics.

## 3. Results

### 3.1. Sociodemographic Characteristics of Study Participants

A total of 8569 pupils were included in our study. Among them, only 3649 have a history of going out. Ages ranged from 6 to 13 years with a median (interquartile range, IQR) of 9 (8–11, 3). The majority of the participants (52.75%) were boys. 33.82% (2898/8569) of the total study participants were in grade 1 or grade 2 and 31.43% (1147/3569) of the students had an out-going history. For father’s occupation, general employees were the most, with 3239 students, accounting for 37.80%. Only 142 students’ fathers are temporary workers or laid-off and unemployed. 3006 pupil’s mothers were general employees and 164 were operators of production and transportation equipment. In the educational background of parents, the majority of the population were junior college or undergraduates, of which 4452 are fathers and 4241 are mothers. Over 80% of the participants were in Wuhan. More than half (51.95%) of the students filled in the time (one month backward) was before Wuhan locked-down (see Table 1 for details).

### 3.2. Group Comparison in Mask-Wearing Behavior and Hand Hygiene

The chi-square test results are shown in Table 1. Hand hygiene was statistically significant for gender (*χ*^2^ = 6.580, *p* > 0.05), grade (*χ*^2^ = 115.186, *p* > 0.01), out-going history (*χ*^2^ = 99.787, *p* > 0.01), father’s occupation (*χ*^2^ = 20.155, *p* > 0.01), mother’s occupation (*χ*^2^ = 16.759, *p* > 0.05), mother’s educational background (*χ*^2^ = 11.597, *p* > 0.01) and time filling out the survey (*χ*^2^ = 36.412, *p* > 0.01). For mask-wearing behavior, statistical significance was observed for grade (*χ*^2^ = 6.629, *p* > 0.05), father’s occupation (*χ*^2^ = 35.378, *p* > 0.01), mother’s occupation (*χ*^2^ = 44.877, *p* > 0.01), father’s educational background (*χ*^2^ = 119.503, *p* > 0.01), mother’s educational background (*χ*^2^ = 123.880, *p* > 0.01), residence (*χ*^2^ = 19.151, *p* > 0.01) and time filling out the survey (*χ*^2^ = 23.261, *p* > 0.01).

### 3.3. Factors Associated with Hand Hygiene and Mask-Wearing Behavior

In the binary analysis, gender, grade, out-going history, father’s occupation, mother’s educational background and time filling out the survey were significantly associated with better hand hygiene (Table 2). The OR of girls was 1.12 times higher (95% CI = 1.03–1.22, *p* < 0.05) compared to boys. Students in grade 3 and 4 were 1.25 times (95% CI = 1.13–1.40, *p* < 0.01) and students in grade 5 and 6 were 1.72 times (OR = 1.72, 95% CI = 1.55–1.92, *p* < 0.01) as much as students in grade 1 and 2. Study participants who reported out-going history were more likely to report better hand hygiene than those who had no out-going history with an estimated OR of 1.51 (95% CI = 1.38–1.65, *p* < 0.01). Compared to fathers who were temporary workers or laid-off and unemployed, all occupations were statistically significant. Professional and technical personnel, general employees, managers of state agencies, enterprises and institutions, business and service workers, peasant, operators of production and transportation equipment and others, separately with the OR of 1.49 (95% CI = 1.02–2.18, *p* < 0.05), 1.48 (95% CI = 1.03–2.13, *p* < 0.05), 1.89 (95% CI = 1.25–2.86, *p* < 0.01), 1.78 (95% CI = 1.22–2.58, *p* < 0.01), 1.66 (95% CI = 1.06–2.60, *p* < 0.05), 1.58 (95% CI = 1.06–2.34, *p* < 0.05) and 1.57 (95% CI = 1.09–2.28, *p* < 0.05). Time filling out the survey with after Wuhan locked-down was 0.76 times (95% CI = 0.69–0.83, *p* < 0.01) than that of students before Wuhan locked-down The results of the binary logistic analysis showed that grade, mother’s educational background, and residence were significantly associated with better mask-wearing behavior (Table 3). The results from multivariate logistic regression analysis revealed the OR of students in grade 5 and 6 were 1.21 times higher (95% CI = 1.03–1.43, *p* < 0.05) compared to students in grade 1 and 2. Mother’s educational background with junior college or undergraduate (OR = 1.87, 95% CI = 1.05–3.33, *p* < 0.05) and postgraduate and above (OR = 2.28, 95% CI = 1.12–4.65, *p* < 0.05) was more likely to report better mask-wearing behavior than those educational background with primary school and below. Students who living in Hubei (except for Wuhan) (OR = 0.70, 95% CI = 0.55~0.88, *p* < 0.01) and outside Hubei (OR = 0.76, 95% CI = 0.62–0.93, *p* < 0.01) was less likely to report better behavior.

## 4. Discussion

Hand hygiene is considered a remarkably important element of infection control. Previous studies have confirmed the effect hand-washing imposed on the prevalence of respiratory illness [11,12], claiming that an appropriate hand-washing intervention could break the transmission cycle and reduce the risk between 6% and 44% [13]. Although hand-washing is recommended as an inexpensive and widely available protective measure for both personal protection and epidemic prevention of some viral respiratory infections, like influenza and severe acute respiratory syndrome [14,15]^,^ it is very difficult to maintain high hand-washing compliance. Owing to different populations and contexts, the compliance of hand-washing ranges from 1.80% to 78.00% [16,17]. As can be seen in our study, only 42.05% (3603/8569) of primary school children showed an excellent hand-washing cognition and behavior, far below our expectation. Through further analysis of children’s hand-washing behavior and its impact factors, it was found to need improvement in the following aspects:

The first aspect is to strengthen the supervision of hand-washing among schoolboys, because girls showed 1.12 times the possibility of excellent hand-washing compared with boys in this study. Consistent with our findings, several previous studies have revealed an obvious gender distinction regarding the perception, behavior, and effectiveness of hand-washing [18,19] About this difference, it has been postulated that females are less willing to take risky activities, and thus more likely to follow hand-washing recommendations [20]. Second, hand hygiene education is remarkably beneficial in preventing infectious diseases, especially for young children in primary schools. It was found that children’s behavior of hand-washing was closely related to their grade, fathers’ occupation in this study. Accordingly, family hygiene education should be conducted based on children’s cognitive capacity. Parents must raise the level of self-protection awareness, make themselves disciplined, and improve the guidance to children to make up the negligence of education and the negative influence due to the possible lower-rank profession. It is necessary to make full use of the quarantine period to help children developing a good personal hygienic awareness and instructing children to keep a correct hand-washing behavior in the form of rewards. Considering better hand hygiene expressed among the students with an outgoing history or before Wuhan locked-down, the last method is to enlarge medium publicity strength in increasing the awareness and frequency of hand-washing even if your city is lockdown or you are required to not leave your residence.

Similar to hand-washing, a strict self-protection measure like mask-wearing is also used to prohibit pathogens entering the respiratory tract by cutting off the droplet transmission route directly. The feasibility and effectiveness have been verified in previous researches [6,7,21]. At present, westerners generally opposed mask-wearing by normal people, but the experience in China and South Korea showed that mask-wearing was an effective measure. For example, N95 masks and surgical masks could separately block 91% and 68% of pathogens [7]. As a susceptible population, a properly fitted face mask plays a crucial importance for the youth, but our study revealed that only 32.47% of the primary school children used properly fitted face masks, whereas 42.42% said it was difficult to buy kids’ masks. Along with China’s resumption of work and a new production standard of children’s masks, the output and the quality of children’s masks would be improved in a short time. And based on a multiple-factors analysis, we observed that higher grade, higher education level of the mother, and the residence of Wuhan are significant factors affecting the behavioral level of mask-wearing.

Concerning the mother’s education level, it may be explained as the increased awareness of perceived susceptibility and severity. As they might have a better understanding of public health measures and their effectiveness, and consequently a higher perception and compliance of the benefits of mask-wearing. It also allows them to strengthen children’s safety education, which would probably result in a higher mask-wearing level in the youth [20]. Similar to those of the aforementioned improvement measures, mask-wearing education should be following children’s cognitive capacity; an enlarged public campaign regarding mask-wearing through mass media and public education would be appreciated even in the areas less hit by COVID-19; parents should pay more attention to behavior guidance.

This study is not only a supplementary of hand-washing and mask-wearing in the past studies, but also contributes to a better understanding of primary school children’s behaviors, which makes a lot of sense to China and other countries. Even so, it has some limitations. The sample size is large, but it is still insufficient. Only 15 schools were included. It is recommended that the government conduct a second round of surveys, which will include all primary schools in Wuhan to further improve the representativeness of the sample. Recall bias and social desirability bias may be caused by the self-reported property of the research. Some unknown and omitted confounding factors may cause residual confounding, instrumental variable analysis were used to control these confounding factors. In addition, a cause-effect relationship cannot be established due to the inherent nature of cross-sectional design.

## 5. Conclusions

Only 42.05% of the pupils showed an excellent hand-washing cognition and behavior. Gender, grade, outgoing history, father’s occupation, mother’s educational background and time filling out the survey were significantly associated with better hand hygiene. 51.60% of the students showed good mask-wearing behavior. Grade, mother’s educational background, and residence were significantly associated with better mask-wearing behavior. Hand hygiene and mask-wearing behavior education are remarkably beneficial in preventing infectious diseases. Parents should pay more attention to behavior guidance. Government can also increase the awareness of students through enlarging medium publicity

## Figures and Tables

**Table 1 ijerph-17-02893-t001:** Comparison of good rate of hand hygiene (*n* = 8569) and mask-wearing (*n* = 3649) among pupils in different groups.

Category	Group	Hand Hygiene			Mask-Wearing		
		Good Rate/Poor Rate (*n*/%)	*χ*^2^	*p*	Good Rate/Poor Rate (*n*/%)	*χ*^2^	*p*
Gender	male	1842 (40.75)/2678 (59.25)	6.580	0.010 *	986 (51.25)/938 (48.75)	0.206	0.650
	female	1761(43.49)/2288 (56.51)			897 (52.00)/828 (48.00)		
Grade	grade 1 and grade 2	1030 (35.54)/1868 (64.46)	115.186	0.001 **	581 (50.65)/566 (49.35)	6.629	0.036 *
	grade 3 and grade 4	1165 (41.25)/1659 (58.75)			599 (49.50)/611 (50.50)		
	grade 5 and grade 6	1408 (49.46)/1439 (50.54)			703 (54.41)/589 (45.59)		
Out-going history	no	1843 (37.46)/3077 (62.54)	99.787	0.000 **	——	——	——
	yes	1760 (48.23)/1889 (51.77)			——		
Father’s occupation	temporary workers or laid-off and unemployed	47 (33.10)/95 (66.90)	20.155	0.005 **	21 (41.18)/30 (58.82)	35.378	0.000 **
	professional and technical personnel (doctors, teachers, lawyers, engineers)	444 (40.62)/649 (59.38)			274 (60.49)/179 (39.51)		
	general employees (general civil servants, company employees)	1306 (40.32)/1933 (59.68)			720 (52.21)/659 (47.79)		
	managers of state agencies, enterprises and institutions	184 (46.70)/210 (53.30)			89 (55.28)/72 (44.72)		
	business and service workers	570 (45.53)/682 (54.47)			280 (50.82)/271 (49.18)		
	peasant	96 (42.29)/131 (57.71)			34 (32.69)/70 (67.31)		
	operators of production and transportation equipment	244 (43.26)/320 (56.74)			125 (49.60)/127 (50.40)		
	others	712 (42.94)/946 (57.06)			340 (48.71)/358 (51.29)		
Mother’s occupation	temporary workers or laid-off and unemployed	137 (39.26)/212 (60.74)	16.759	0.019 *	22 (59.46)/15 (40.54)	44.877	0.000 **
	professional and technical personnel (doctors, teachers, lawyers, engineers)	370 (39.15)/575 (60.85)			250 (60.68)/162 (39.32)		
	general employees (general civil servants, company employees)	1227 (40.82)/1779 (59.18)			682 (54.39)/162 (39.32)		
	managers of state agencies, enterprises and institutions	49 (47.12)/55 (52.88)			329 (50.46)/323 (49.54)		
	business and service workers	639 (44.25)/805 (55.75)			33 (42.86)/44 (57.14)		
	peasant	70 (39.11)/109 (60.89)			65 (43.92)/83 (56.08)		
	operators of production and transportation equipment	61 (37.20)/103 (62.80)			26 (29.89)/61 (70.11)		
	others	1050 (44.15)/1328 (55.85)			476 (48.47)/506 (51.53)		
Father’s educational background	primary school and below	42 (50.00)/42 (50.00)	6.063	0.195	14 (35.00)/26 (65.00)	119.503	0.000 **
	junior high school	530 (44.50)/661 (55.50)			188 (33.45)/374 (66.55)		
	high school/technical secondary school/secondary vocational technical school	1020 (41.85)/1417 (58.15)			555 (50.18)/551 (49.82)		
	junior college/undergraduate	1846 (41.46)/2606 (58.54)			1009 (57.01)/761(42.99)		
	Postgraduate and above	165 (40.74)/240 (59.26)			117 (68.42)/54 (31.58)		
Mother’s educational background	primary school and below	71 (46.41)/82 (53.59)	11.597	0.021 *	22 (34.38)/42 (65.62)	123.880	0.000 **
	junior high school	580 (41.49)/802 (58.03)			250 (37.20)/422 (62.80)		
	high school/technical secondary school/secondary vocational technical school	1128 (44.46)/1409 (55.54)			531 (47.71)/582 (52.29)		
	junior college/undergraduate	1724 (40.65)/2517 (59.35)			999 (59.46)/681 (40.54)		
	Postgraduate and above	100 (39.06)/156 (60.94)			81 (67.50)/39 (32.50)		
Residence	Wuhan	2974 (41.74)/4151 (58.26)	5.380	0.068	1529 (53.48)/1330 (46.52)	19.151	0.000 **
	All cities in Hubei Province except for Wuhan	272 (40.84)/394 (59.16)			146 (43.32)/191 (56.68)		
	All provinces except Hubei	357 (45.89)/421 (54.11)			208 (45.92)/245 (54.08)		
Time filling out the survey (one month backward)	Before Wuhan locked-down	2402 (44.51)/2994 (55.49)	36.412	0.000 **	1298 (54.51)/1083 (45.49)	23.261	0.000 **
	After Wuhan locked-down	1201 (37.85)/1972 (62.15)			585 (46.14)/683 (53.86)		
Total	——	360 (42.05)/4966 (57.95)	——	——	1883 (51.60)/1766 (48.40)	——	——

Notes: * *p* < 0.05, ** *p* < 0.01.

**Table 2 ijerph-17-02893-t002:** Binary logistic regression analysis on the influencing factors of hand hygiene (*n* = 8569).

Variables	β	S.E.	Wald	*p*	OR	OR 95% CI
**Gender**						
male					1.00	
female	0.11	0.05	6.34	0.012 *	1.12	1.03–1.22
**Grade**						
grade 1 and grade 2					1.000	
grade 3 and grade 4	0.23	0.06	16.83	0.000 **	1.25	1.13–1.40
grade 5 and grade 6	0.54	0.06	98.90	0.000 **	1.72	1.55–1.92
**Out-going history**						
no					1.00	
yes	0.41	0.05	83.64	0.000 **	1.51	1.38–1.65
**Father’s occupation**						
temporary workers or laid-off and unemployed					1.00	
professional and technical personnel (doctors, teachers, lawyers, engineers)	0.40	0.19	4.25	0.039 *	1.49	1.02–2.18
general employees (general civil servants, company employees)	0.39	0.19	4.42	0.035 *	1.48	1.03–2.13
managers of state agencies, enterprises and institutions	0.64	0.21	9.23	0.002 **	1.89	1.25–2.86
business and service workers	0.58	0.19	9.09	0.003 **	1.78	1.22–2.58
peasant	0.51	0.23	4.93	0.026 *	1.66	1.06–2.60
operators of production and transportation equipment	0.46	0.20	5.15	0.023 *	1.58	1.06–2.34
others	0.45	0.19	5.80	0.016 *	1.57	1.09–2.28
**Mother’s educational background**						
primary school and below					1.00	
junior high school	−0.19	0.17	1.12	0.27	0.83	0.59–1.16
high school/technical secondary school/secondary vocational technical school	−0.10	0.17	0.30	0.58	0.91	0.65–1.27
junior college/undergraduate	−0.25	0.17	2.18	0.14	0.78	0.56–1.09
Postgraduate and above	−0.38	0.22	3.16	0.08	0.68	0.45–1.04
**Time filling out the survey (one month backward)**						
Before Wuhan locked-down					1.00	
After Wuhan locked-down	−0.28	0.05	34.08	0.000 **	0.76	0.69–0.83
**Constant**	−0.96	0.24	15.40	0.000 **	0.38	

Notes: S.E.—standard error, OR—odds ratio, 95% CI—95% confidence interval, * *p* < 0.05, ** *p* < 0.01.

**Table 3 ijerph-17-02893-t003:** Binary logistic regression analysis on the influencing factors of mask-wearing behavior (*n* = 3649).

Variables	β	S.E.	Wald	*p*	OR	OR 95% CI
**Grade**						
grade 1 and grade 2					1.00	
grade 3 and grade 4	−0.02	0.09	0.07	0.79	0.98	0.83–1.15
grade 5 and grade 6	0.19	0.08	5.17	0.023 *	1.21	1.03–1.43
**Father’s educational background**						
primary school and below					1.00	
junior high school	−0.13	0.36	0.13	0.72	0.88	0.43–1.77
high school/technical secondary school/secondary vocational technical school	0.40	0.36	1.25	0.26	1.49	0.74–3.02
junior college/undergraduate	0.42	0.36	0.13	0.25	1.52	0.75–3.10
Postgraduate and above	0.75	0.41	3.42	0.065	2.11	0.96–4.68
**Mother’s educational background**						
primary school and below					1.00	
junior high school	0.05	0.29	0.03	0.86	1.05	0.60–1.86
high school/technical secondary school/secondary vocational technical school	0.20	0.29	0.48	0.49	1.22	0.69–2.17
junior college/undergraduate	0.62	0.30	4.45	0.035 *	1.87	1.05–3.33
Postgraduate and above	0.82	0.36	5.14	0.023*	2.28	1.12–4.65
**Residence**						
Wuhan					1.000	
All cities in Hubei Province except for Wuhan	−0.36	0.12	9.15	0.002 **	0.70	0.55–0.88
All provinces except Hubei	−0.27	0.10	6.87	0.009 **	0.76	0.62–0.93
**Constant**	−0.66	0.38	2.90	0.088	0.52	——

Notes: * *p* < 0.05, ** *p* < 0.01.

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
