# Peer review of "Hand Hygiene, Mask-Wearing Behaviors and Its Associated Factors during the COVID-19 Epidemic: A Cross-Sectional Study among Primary School Students in Wuhan, China"

_ijerph, 2020, doi:10.3390/ijerph17082893_

Round 1

Reviewer 1 Report

In this paper the authors described the results of a study regarding the behaviors of hand-washing and mask-wearing among primary school students in Wuhan, China. The obtained results underlined the importance of several socio-demographic characteristics (gender, grade and others) and the role of parents on improving the correct behaviors.

The results are very interesting relating to the present COVID 19 pandemic situation and they could be used as basis for increase the interventions for better inform people of the crucial role of hand washing and mask using in this phase.

In my opinion the introduction must be improved with more recent bibliographical resources, for example, I suggest the following reference: Ma QX et al. Potential utilities of mask-wearing and instant hand hygiene for fighting SARS-CoV-2. Med Virol. 2020 Mar 31. Doi: 10.1002 / jmv.25805.

To clarify the results, I suggest to insert only the more significant data in tables or, better, by creating figures. The authors could be add the other data in supplementary materials section.

The authors could better explain what are the specific suggestions for parents to increase the good behavior in students.

The manuscript could be published in the journal after minor revisions.

Author Response

Reviewer 1

Open Review

English language and style

( ) Extensive editing of English language and style required 
( ) Moderate English changes required 
(x) English language and style are fine/minor spell check required 
( ) I don't feel qualified to judge about the English language and style 

Yes
Can be improved
Must be improved
Not applicable
Does the introduction provide sufficient background and include all relevant references?
( )
(x)
( )
( )
Is the research design appropriate?
(x)
( )
( )
( )
Are the methods adequately described?
(x)
( )
( )
( )
Are the results clearly presented?
(x)
( )
( )
( )
Are the conclusions supported by the results?
(x)
( )
( )
( )
Comments and Suggestions for Authors

In this paper the authors described the results of a study regarding the behaviors of hand-washing and mask-wearing among primary school students in Wuhan, China. The obtained results underlined the importance of several socio-demographic characteristics (gender, grade and others) and the role of parents on improving the correct behaviors.

The results are very interesting relating to the present COVID 19 pandemic situation and they could be used as basis for increase the interventions for better inform people of the crucial role of hand washing and mask using in this phase.

In my opinion the introduction must be improved with more recent bibliographical resources, for example, I suggest the following reference: Ma QX et al. Potential utilities of mask-wearing and instant hand hygiene for fighting SARS-CoV-2. Med Virol. 2020 Mar 31. Doi: 10.1002 / jmv.25805.

Answer: Thanks for your suggestion. We have read this article carefully, and the research is really meaningful and interesting. This research can help us to strengthen the credibility of our views, so we want to cite this article.

To clarify the results, I suggest to insert only the more significant data in tables or, better, by creating figures. The authors could be add the other data in supplementary materials section.

Answer: Thank you for your suggestion. We have modified tables.

The authors could better explain what are the specific suggestions for parents to increase the good behavior in students.

Answer: Thank you for your advice. As is mentioned in the manuscript, parents should do from two aspects: learn more and play a leading role in developing a hygienic awareness and behavior; teach children based on their cognitive capacity. We also added a tip of rewards in the revised manuscript to make this process more interesting.

The manuscript could be published in the journal after minor revisions.

Thanks again for your earnest work. Your suggestions are helpful to improve this paper. We sincerely hope the correction will meet your approval.

Reviewer 2 Report

The authors have done a great work of exploring the various factors and their significance to hand washing and mask wearing practices in Wuhan. The results that were presented given different variables were very useful and interesting along with recommendations that were provided to increase the level of parent’s self-awareness and guidance as well as the government’s role.

The authors mentioned the questionnaire and few questions, but it would also be great to get a little bit more detail about this survey. How many questions in total? How many parts did it have? Is this available to view online? Any reference for it?

They mentioned limitations to this research, is there any other downside or limitations? Any future directions or suggestions?

The paper is a well-organized paper and is written in a logical order. However, I would strongly recommend to re-edit the paper. There are many grammatical errors. I can point to few:

Line 13: The beginning sentence of the abstract which is very important in the paper since readers start the article with the abstract: “While the behaviors of hand-washing and mask-wearing are repeatedly emphasized during the pandemic of Coronavirus Disease 2019, not all people attach enough importance to.” (Consider revising or editing! This does not read correctly. )

Line 40: In China, wearing masks are also necessary. Hand-washing is suitable for all people, but keep the social distance is only suitable for adults when they go out. (keeping)

Line 51: Nevertheless, primary school children, 51 as a type of susceptible population isolated at home, whose personal protection is often overlooked. (Consider editing or revising)

Line 59: The cross-sectional and quantitative study is conducted from Feb 16, 2020 to Feb 25, 2020. A total of 9145 students from 15 primary school in Wuhan, China participates in the survey. (participated)

Author Response

Reviewer 2

Open Review

English language and style

(x) Extensive editing of English language and style required 
( ) Moderate English changes required 
( ) English language and style are fine/minor spell check required 
( ) I don't feel qualified to judge about the English language and style 

Yes

Can be improved

Must be improved

Not applicable

Does the introduction provide sufficient background and include all relevant references?

(x)

( )

( )

( )

Is the research design appropriate?

( )

(x)

( )

( )

Are the methods adequately described?

( )

(x)

( )

( )

Are the results clearly presented?

(x)

( )

( )

( )

Are the conclusions supported by the results?

(x)

( )

( )

( )

Comments and Suggestions for Authors

The authors have done a great work of exploring the various factors and their significance to hand washing and mask wearing practices in Wuhan. The results that were presented given different variables were very useful and interesting along with recommendations that were provided to increase the level of parent’s self-awareness and guidance as well as the government’s role.

The authors mentioned the questionnaire and few questions, but it would also be great to get a little bit more detail about this survey. How many questions in total? How many parts did it have? Is this available to view online? Any reference for it?

Answer: Thank you for your kindly reminding. We provided a copy of the questionnaire as a supplement to get a detail about this survey. The survey included three parts, namely the sociodemographic characteristics (12 questions), hand hygiene (3 questions), and mask-wearing (7 questions).

They mentioned limitations to this research, is there any other downside or limitations? Any future directions or suggestions?

Answer: The sample size is large, but it is still insufficient. Only 15 schools were included. It is recommended that the government conduct a second round of surveys, which will include all primary schools in Wuhan to further improve the representativeness of the sample. Some unknown and omitted confounding factors may cause residual confounding, instrumental variable analysis were used to control these confounding factors.

The paper is a well-organized paper and is written in a logical order. However, I would strongly recommend to re-edit the paper. There are many grammatical errors. I can point to few:

Line 13: The beginning sentence of the abstract which is very important in the paper since readers start the article with the abstract: “While the behaviors of hand-washing and mask-wearing are repeatedly emphasized during the pandemic of Coronavirus Disease 2019, not all people attach enough importance to.” (Consider revising or editing! This does not read correctly. )

Answer: Thanks for your reminding. We replaced “While the behaviors of hand-washing and mask-wearing are repeatedly emphasized during the pandemic of Coronavirus Disease 2019, not all people attach enough importance to” with “Although the emphasis on behaviors of hand-washing and mask-wearing was repeated during the pandemic of Coronavirus Disease 2019 (COVID-19), not everyone paid enough attention to this”.

Line 40: In China, wearing masks are also necessary. Hand-washing is suitable for all people, but keep the social distance is only suitable for adults when they go out. (keeping)

Answer: Thanks for your kindly reminding. We have revised it.

Line 51: Nevertheless, primary school children, 51 as a type of susceptible population isolated at home, whose personal protection is often overlooked. (Consider editing or revising)

Answer: Thank you for your suggestion. We have revised it.

Line 59: The cross-sectional and quantitative study is conducted from Feb 16, 2020 to Feb 25, 2020. A total of 9145 students from 15 primary school in Wuhan, China participates in the survey. (participated)

Answer: Thanks for your kindly reminding. We have revised it.

Thanks again. Your comments and suggestions are valuable for us to improve the quality of this paper. With your help, we improved our manuscript under the circumstance that these changes will not influence the content of the paper, and we hope the correction will meet your approval.